# Ratoon Rice Cropping Mitigates the Greenhouse Effect by Reducing CH_4_ Emissions through Reduction of Biomass during the Ratoon Season

**DOI:** 10.3390/plants12193354

**Published:** 2023-09-22

**Authors:** Xiaojian Ren, Kehui Cui, Zhiming Deng, Kaiyan Han, Yuxuan Peng, Jiyong Zhou, Zhongbing Zhai, Jianliang Huang, Shaobing Peng

**Affiliations:** 1National Key Laboratory of Crop Genetic Improvement, Key Laboratory of Corp Ecophysiology and Farming System in the Middle Reaches of the Yangtze River, Ministry of Agriculture and Rural Affairs, College of Plant Science and Technology of Huazhong Agricultural University, Wuhan 430070, China; 2Wuxue Agro-Technology Extension Service Center, Wuxue 435499, China

**Keywords:** ratoon rice, methane emissions, global warming potential, greenhouse gas intensity, net ecosystem economic benefits

## Abstract

The ratoon rice cropping system (RR) is developing rapidly in China due to its comparable annual yield and lower agricultural and labor inputs than the double rice cropping system (DR). Here, to further compare the greenhouse effects of RR and DR, a two-year field experiment was carried out in Hubei Province, central China. The ratoon season showed significantly lower cumulative CH_4_ emissions than the main season of RR, the early season and late season of DR. RR led to significantly lower annual cumulative CH_4_ emissions, but no significant difference in cumulative annual N_2_O emissions compared with DR. In RR, the main and ratoon seasons had significantly higher and lower grain yields than the early and late seasons of DR, respectively, resulting in comparable annual grain yields between the two systems. In addition, the ratoon season had significantly lower global warming potential (GWP) and greenhouse gas intensity-based grain yield (GHGI) than the main and late seasons. The annual GWP and GHGI of RR were significantly lower than those of DR. In general, the differences in annual CH_4_ emissions, GWP, and GHGI could be primarily attributed to the differences between the ratoon season and the late season. Moreover, GWP and GHGI exhibited significant positive correlations with cumulative emissions of CH_4_ rather than N_2_O. The leaf area index (LAI) and biomass accumulation in the ratoon season were significantly lower than those in the main season and late season, and CH_4_ emissions, GWP, and GHGI showed significant positive correlations with LAI, biomass accumulation and grain yield in the ratoon and late season. Finally, RR had significantly higher net ecosystem economic benefits (NEEB) than DR. Overall, this study indicates that RR is a green cropping system with lower annual CH_4_ emissions, GWP, and GHGI as well as higher NEEB.

## 1. Introduction

Global warming is commonly considered a consequence of increases in greenhouse gases (GHG), such as methane (CH_4_), nitrous oxide (N_2_O), and carbon dioxide (CO_2_). Although the concentration of CH_4_ and N_2_O in the atmosphere is lower than that of CO_2_, their greenhouse effects are 34 and 298 times greater than that of CO_2_ on a 100-year scale [1]. Agricultural activities are primary sources of carbon emissions, contributing 18 Gt CO_2_-eq of GHGs annually, which accounts for 34% of the global anthropogenic GHG emissions [2]. Paddy fields are one of the major contributors of CH_4_ and N_2_O emissions, which account for approximately 48% and 50% of agricultural CH_4_ and N_2_O emissions, respectively [3,4]. It is a great challenge to reduce CH_4_ and N_2_O emissions from paddy fields and at the same time ensure high and stable rice yield in agricultural production for increasing human needs.

It has been reported that approximately 90% of paddy CH_4_ emissions are derived from rice plants [5], and therefore the characteristics of rice plants have a great impact on CH_4_ emissions from paddy fields. For example, the leaf area, plant height, tiller number, and biomass of rice plants are positively associated with CH_4_ emissions [6,7,8,9]. In addition, rice root exudates may promote CH_4_ emissions by providing substrates for methanogens in paddy soils [10,11]. Root radial oxygen loss of rice plants can enhance the oxidation of CH_4_ produced in soils and thus reduce CH_4_ emissions [12]. Plant photosynthates are primary components of root exudates, and increasing the distribution of photosynthetic products in spikelets may improve rice yield and reduce CH_4_ emissions [13,14]. Previous studies have also concluded that rice production is a process to pursue high yield and low GHG emissions [15], and planting of high-yield varieties was found to significantly reduce CH_4_ emissions in China [16,17]. In addition, agricultural management has great impacts on paddy GHG emissions, and fertilizer (particularly nitrogen fertilizer) application and tillage can increase CH_4_ and N_2_O emissions from paddy fields [18,19,20]. Therefore, the selection of appropriate rice varieties and agricultural management in rice cropping systems may enhance the grain yield and lower agricultural GHG emissions [21,22,23].

Global warming potential (GWP) and greenhouse gas intensity based on grain yields (GHGI) are the main indicators to evaluate the greenhouse effects of different cropping systems [24]. Net ecosystem economic benefit (NEEB) is calculated based on the economic income of crop yields, agricultural inputs, and GHG emissions, which considers both the economic benefits and greenhouse effects for a comprehensive evaluation of cropping systems [25]. Previous studies have explored GWP, GHGI, and NEEB in different cropping systems. For example, the garlic-rice system was found to have significantly higher GWP but significantly lower GHGI than the wheat-rice system [26]. The ratoon rice cropping system (RR) showed similar direct GWP to the rice-wheat cropping system, which was 33% lower than that of the double rice cropping system (DR) [27]. These studies suggest that the different rotated crops in a given rice-based cropping system may affect the GWP and GHGI of the system. Compared with single rice with continuous flooding, ratoon rice grown under plastic film mulching showed significant increases in NEEB and dramatic decreases in GHGI [24]. Therefore, management practice also determines the GHGI and NEEB of a given cropping system. Recently, Zhou et al. [28] reported that RR has significantly lower GWP and GHGI than DR in the middle and lower reaches of the Yangtze River in China, and significantly higher GWP and lower GHGI than the middle-season rice system, as well as higher net economic income than the other two systems. So RR may be an environment-friendly and sustainable cropping system. Overall, GWP, GHGI, and NEEB vary considerably among different cropping systems, and optimization of cropping systems can improve economic benefits while reducing environmental costs.

DR and RR are two important cropping systems to ensure stable rice yields in China owing to their capacity to produce grains in two seasons [29,30]. DR is a cropping system comprising the early season and the late season. RR refers to the production of the second rice crop from the stubbles after the harvest of the main crop through appropriate management practices [31]. RR and DR require different management practices (such as tillage and water management) and agricultural inputs (such as seeds, fertilizers, pesticides, and labor), which may lead to differences in GHG emissions, as well as in yield performance and economic benefits [30]. Previous studies have demonstrated that RR generally has lower annual GWP, GHGI, and carbon footprint [27], and higher NEEB compared with DR [28,30]. In addition, Xu et al. [25] have comprehensively studied the GHG emissions, carbon footprint, and NEEB of RR and DR, which has greatly improved our understanding of the GHG emissions and NEEB of the two cropping systems. However, in most previous studies, the GHG emissions were estimated using emission factors, which could not truly reveal the characteristics of GHG emissions from paddy fields [28,30]. Although Zhou et al. [27] directly collected and measured GHG emissions, they did not compare the seasonal GHG emissions between DR and RR. In addition, the accumulation and distribution of photosynthetic substances significantly affect CH_4_ emissions [13]. However, there has been little research on the effects of aboveground plant characteristics on GHG emissions in DR and RR [3,25].

Here, we carried out a two-year field experiment to compare the GHG emissions, GWP, GHGI, and NEEB between RR and DR, and reveal the effects of aboveground characteristics on GHG emissions in the middle reach of the Yangtze River in China, where RR and DR are widely adopted, aiming to provide a better theoretical basis for the development of sustainable RR.

## 2. Results

### 2.1. Grain Yields and Characteristics of Aboveground Rice Plants

The leaf area index (LAI), biomass accumulation from the full heading stage to maturity stage (BAH), and plant aboveground biomass at maturity stage (Biomass) of the main season in RR were higher than those of the early season in DR in 2018 and 2019 (Table 1), but the difference was not significant for BAH. The grain yield of the main season (an average of 9.27 t ha^−1^) was significantly higher than that of the early season (an average of 7.02 t ha^−1^) in both years (Table 1).

In terms of the second season, the ratoon season had significantly lower LAI and biomass than the late season (Table 1). In addition, the BAH of the ratoon season was significantly lower in 2018 but significantly higher in 2019 than that of the late season, and when averaged across two years, the ratoon season had a significantly lower BAH than the late season. The grain yield in the ratoon season was 5.55 t ha^−1^ and 6.73 t ha^−1^, and that in the late season was 8.64 t ha^−1^ and 9.64 t ha^−1^ in 2018 and 2019, respectively (Table 1).

### 2.2. CH_4_ and N_2_O Emissions

The main season in RR had higher cumulative CH_4_ emissions than the early season in DR, but the difference was not significant in 2019 (Table 2). The average cumulative CH_4_ emissions across the two years were 44.43 g m^−2^ in the main season, which was not significantly different from that in the early season (36.67 g m^−2^).

The cumulative CH_4_ emissions in the ratoon season were 35.42% of those in the main season in 2018 and 30.49% of those in the main season in 2019. Additionally, the cumulative CH_4_ emissions in the ratoon season were 14.52 g m^−2^ in 2018 and 14.59 g m^−2^ in 2019, which were significantly lower than those in the late season (64.83 g m^−2^ and 43.92 g m^−2^) (Table 2). The annual cumulative CH_4_ emissions from RR were 55.53 g m^−2^ in 2018 and 62.45 g m^−2^ in 2019, which were significantly lower than those from DR (94.86 g m^−2^ and 87.23 g m^−2^, respectively).

Compared with those of the early season, the cumulative N_2_O emissions of the main season were similar in 2018, but significantly lower in 2019 (Table 2); besides, when averaged across two years, the annual cumulative N_2_O emissions were not significantly different between the main season (94.13 mg m^−2^) and early season (81.62 mg m^−2^). In addition, there was no significant difference in cumulative N_2_O emissions between the ratoon season and late season in both years (Table 2). The annual cumulative N_2_O emissions were 329.32 and 155.70 mg m^−2^ in RR, and 322.58 and 170.81 mg m^−2^ in DR in 2018 and 2019, respectively, showing no significant difference between RR and DR (Table 2).

### 2.3. GWP and GHGI

The GWP of the main season was higher than that of the early season, and the difference was significant in 2018 but not in 2019 (Table 3). However, for average GWP across two years, there was no significant difference between the main season and the early season (Table 3). The GWP was 5.49 and 5.30 t CO_2_-eq ha^−1^ in the ratoon season, and 22.68 and 15.28 t CO_2_-eq ha^−1^ in the late season in 2018 and 2019, respectively (Table 3). RR had an annual GWP of 19.86 t CO_2_-eq ha^−1^ in 2018 and 21.70 t CO_2_-eq ha^−1^ in 2019, which were significantly lower than those of DR (33.21 t CO_2_-eq ha^−1^ in 2018 and 30.17 t CO_2_-eq ha^−1^ in 2019). Similarly, RR had a significantly lower average annual GWP across the two years (20.78 t CO_2_-eq ha^−1^) than DR (31.69 t CO_2_-eq ha^−1^).

There was no significant difference in GHGI between the main season and early season in the two years (Table 3). However, the GHGI of the ratoon season was only 37.93% and 50.12% that of the late season in 2018 and 2019, respectively (Table 3). In addition, the average GHGI across two years was 0.89 t CO_2_-eq t^−1^ yield in the ratoon season, which was significantly lower than that in the late season (2.10 t CO_2_-eq t^−1^ yield). The annual GHGI of RR was 1.22 t CO_2_-eq t^−1^ yield in 2018 and 1.49 t CO_2_-eq t^−1^ yield in 2019, which was significantly lower than that of DR (2.07 t CO_2_-eq t^−1^ yield in 2018 and 1.85 t CO_2_-eq t^−1^ yield in 2019). The average annual GHGI of RR (1.36 t CO_2_-eq t^−1^ yield) across two years was also significantly lower than that of DR (1.96 t CO_2_-eq t^−1^ yield).

### 2.4. Correlations among GHG Parameters and the Aboveground Characteristics of Crops

GWP showed significant positive correlations with cumulative CH_4_ emissions (Figure 1A), but no correlation with cumulative N_2_O emissions (Figure 1B). GHGI was significantly positively correlated with cumulative CH_4_ emissions (Figure 1C) but showed no correlation with cumulative N_2_O emissions (Figure 1D).

In the first season, cumulative CH_4_ emissions and GWP exhibited no significant correlation with biomass, LAI, BAH, and grain yield (Table 4). GHGI was not correlated with LAI, BAH, and biomass, but had a significant negative correlation with grain yield. In the second season, cumulative CH_4_ emissions, GWP, and GHGI were all significantly positively correlated with LAI, BAH, biomass, and grain yield (Table 4).

### 2.5. Economic Benefits, Total Costs, and NEEB

In the first season, the grain income in the main season of RR was significantly higher than that in the early season of DR. However, there was no significant difference in total costs between the main season and the early season (Table 5). In the second season, the grain income and total costs in the ratoon season of RR were significantly lower than those in the late season of DR. When averaged across two years, RR has significantly lower annual grain income than DR (43,457 vs. 46,384 CNY ha^−1^), as well as significantly lower total costs (17,420 vs. 27,794 CNY ha^−1^) (Table 5).

In the first season, the main season had significantly higher NEEB than the early season in both years (Table 5). Similarly, the ratoon season exhibited higher NEEB than the late season, but the difference was not significant in 2019. When averaged across two years, the ratoon season had higher NEEB than the late season (14,440 vs. 12,734 CNY ha^−1^), and RR resulted in higher annual NEEB than DR (26,037 vs. 18,590 CNY ha^−1^).

## 3. Discussion

### 3.1. Differences in GHG Emissions between Ratoon Rice Cropping System and Double Rice Cropping System

In this study, RR had significantly lower annual cumulative CH_4_ emissions than DR (Table 2), which is consistent with the findings of Zhou et al. [27]. The main season had similar cumulative CH_4_ emissions to the early season; however, the ratoon season had significantly lower cumulative CH_4_ emissions than the late season (Table 2), which is consistent with the results reported by Xu et al. [25]. Therefore, the lower annual cumulative CH_4_ emissions in RR relative to DR are primarily attributed to the lower CH_4_ emissions in the ratoon season than those in the late season.

Furthermore, the average CH_4_ emission flux in the ratoon season was significantly lower than that in the late season (Appendix A). Rice plants can affect paddy CH_4_ emissions by regulating the distribution of photosynthates among aboveground and underground parts of plants [14]. A higher LAI is often conducive to the interception of radiation for photosynthesis [32,33]. Low LAI and biomass generally result in lower CH_4_ emissions due to lower production of root exudates [6,11]. The LAI and biomass accumulation were significantly lower in the ratoon season than in the late season (Table 1), and there was a significant positive correlation between aboveground characteristics and CH_4_ emissions (Table 4). Huang [34] also found that lower leaf area and lower the ratio of leaf area to number of spikelets resulted in lower CH_4_ emissions, and the carbon content in root exudates has a positive relation with CH_4_ emissions. These results imply that less photosynthetic carbon may be allocated to the roots and soils through root exudates in the ratoon season, thereby reducing the substrates for methanogens and then decreasing CH_4_ emissions [11]. Moreover, compared with conventional or reduced tillage, no-tillage can significantly reduce CH_4_ emissions [35]. In this study, conventional tillage was conducted for the late season but no-tillage was carried out for the ratoon season, which might be responsible for the lower CH_4_ emissions in the ratoon season. Low nitrogen application was also found to significantly reduce CH_4_ emissions in the early season and late season [18]. The actual application rate of nitrogen fertilizer was 75 kg ha^−1^ as tiller-promoting fertilizer when measuring the GHG emissions during the ratoon season, which was much lower than the nitrogen application (150 kg ha^−1^) in the late season. Therefore, the difference in CH_4_ emissions between the ratoon season and the late season can be ascribed to the differences in aboveground biomass accumulation and agricultural management. Future studies should be carried out to explore the possible physiological reasons for the lower CH_4_ emissions in the ratoon season in terms of grain yield formation and root exudates.

The annual cumulative N_2_O emissions ranged from 155.70 to 329.32 mg m^−2^ in RR and from 170.81 to 322.58 mg m^−2^ in DR (Table 2), which are generally within the range reported by previous studies in RR [36] and DR [37]. Our results revealed that there was no significant difference in cumulative N_2_O emissions between RR and DR (Table 2), which may be attributed to the similar annual growth duration (Table 6) and average N_2_O emission flux between the two cropping systems (Appendix A). In addition, soil moisture has great impacts on nitrification and denitrification associated with N_2_O production [38]. In this study, annual precipitation during rice growth was similar between RR and DR (Table 6). The water management in the late season of DR was somewhat different from that in the ratoon season of RR. However, it is difficult to clarify the effect of differences in irrigation on cumulative N_2_O emissions in the study.

In this study, the ratoon season had significantly higher cumulative N_2_O emissions and average N_2_O emission flux than the main season (Table 2, Appendix A). As mentioned previously, the left stubbles in the ratoon season may enhance root oxygen loss, providing suitable oxygen-rich conditions for the production of N_2_O and thereby increasing N_2_O emissions [39]. Similarly, cumulative N_2_O emissions in the late season were significantly higher than those in the early season (Table 2). Straw incorporation can also increase N_2_O emissions [40]. In addition, precipitation was lower in the second season than in the first season in this study (Table 6), resulting in relatively lower soil moisture and a corresponding increase in soil redox potential [41], which finally increased N_2_O production and emissions [42]. Therefore, the higher N_2_O emissions in the second season should be primarily attributed to the left stubbles of the main season, straw returning to the late season, and less precipitation in the study.

### 3.2. Differences in Methane Emissions between the Main and Ratoon Seasons in Ratoon Rice Cropping System

Our results showed that the cumulative CH_4_ emissions in the ratoon season were 64.04% lower than those in the main season of RR, accounting for 26.45% of the annual cumulative CH_4_ emissions (Table 2). Similarly, cumulative CH_4_ emissions in the ratoon season were found to account for 6.37–35.24% of the annual CH_4_ emissions in previous reports [24,36,43]. Ratoon crops generally have shorter growth periods than main crops [44]. In this study, the growth duration of the ratoon crop was on average about 40 days shorter than that of the main crop (Table 6). Moreover, the ratoon season had a significantly lower average CH_4_ emission flux than the main season (Appendix A). Therefore, the shorter growth duration and lower CH_4_ emission flux together contributed to the lower CH_4_ emissions in the ratoon season.

Several plant characteristics of the ratoon crops may also contribute to the lower CH_4_ emissions in the ratoon season. Firstly, the lower emissions were also partially attributed to the differences in leaf area, biomass accumulation, and root exudates between the main and ratoon seasons, as discussed above. Second, ratoon crops often have lower plant heights than main crops [44], which can reduce the CH_4_ emissions from rice plants [7]. In addition, oxygen enrichment in soils is favorable for the activity of aerobic methanotrophs [45]. After harvest of the main crop, the cut stubbles may make air easily enter the paddy soil via root oxygen loss and therefore enhance oxidization of the produced CH_4_ and inhibit the production of CH_4_ by anaerobic methanogens [45]. Although this study provides evidence for the lower CH_4_ emissions from the ratoon crops in RR, the underlying biological mechanism remains to be explored in the future.

### 3.3. Differences in GWP and GHGI between Ratoon Rice Cropping System and Double Rice Cropping System

This study showed that RR had significantly lower annual GWP and GHGI than DR (Table 3), which is consistent with previous reports [28,30]. In terms of different seasons, there was no significant difference in GWP and GHGI between the main season and early season (Table 3); however, the two parameters in the ratoon season were significantly lower than those in the late season (Table 3). Therefore, the lower annual GWP and GHGI in RR can be primarily attributed to the contribution of the ratoon season.

Our results showed that GWP and GHGI were significantly positively correlated with LAI and biomass accumulation in the second season (Table 4). Hence, the lower LAI and biomass accumulation may partially account for the lower GWP and GHGI in the ratoon season (Table 1 and Table 3). Moreover, we found that cumulative CH_4_ emissions contributed nearly 100% to the GWP (Appendix A) and GHGI (Appendix A). Previous studies have also found that CH_4_ emissions contributed over 98% to the GWP, while N_2_O emissions only accounted for less than 2% of it [46,47]. We also observed that GWP (Figure 1A) and GHGI (Figure 1C) had significant positive correlations with CH_4_ emissions, but no significant correlation with cumulative N_2_O emissions (Figure 1B,D), indicating that the lower cumulative CH_4_ emissions in ratoon season are responsible for the lower annual GWP and GHGI in RR.

### 3.4. Differences in NEEB between the Ratoon Rice Cropping System and Double Rice Cropping System

The NEEB of a given system is often calculated based on the economic income (grain yield and unit price of rice grains), the costs of agricultural inputs (seeds, fertilizers, pesticides, and labor), GWP cost (GWP and carbon-trading price) [24]. Therefore, when estimating the NEEB, several factors should be considered, such as the characteristics of a used variety, farm management, environmental conditions (ambient temperature and rainfall), and so on. In the study, the crop characteristics and CH_4_ emissions were investigated in farmer fields, and the varieties and farm management (seedling raising, transplanting, fertilization, irrigation, pest and disease control, and harvest) in the experiment are widely adopted in the double rice and ratoon rice practice for better annual grain yield and safe production based on the local environments. So, the estimated NEEB in the study should reflect the reality in the experimental site.

The selection of a suitable cropping system is primarily dependent on high net economic benefits, which are mainly determined by the economic income of grain yields and costs of agricultural inputs [21]. In this study, RR had a 6.3% lower annual grain income than DR (Table 5), which could be mainly attributed to the lower grain yield in the ratoon season relative to that in the late season. However, compared with DR, RR significantly decreased the total costs by 37.3% (Table 5), resulting in a higher annual NEEB than that of DR (Table 5). This finding is consistent with previous results [25,28,30].

Our results showed that the difference in total costs between RR and DR could be mainly attributed to differences in the costs of nitrogen fertilizers, seeds, labor, and GWP (Appendix A). In the first season, the same amount of labor was needed, and there was no significant difference in GWP between the two cropping systems (Table 7). Although the nitrogen fertilizer rate was higher in the main season than in the early season, the use of fewer seeds and higher grain yield in the main season may offset the nitrogen fertilizer costs (Table 1 and Table 7). Therefore, the NEEB of the main season was significantly higher than that of the early season (Table 5), which is in agreement with the report of Xu et al. [25]. Compared with the ratoon season, the late season rice production requires seeds and labor for seeding, tilling, and transplanting [22]. In addition, the lower GWP in the ratoon season led to lower total cost relative to the late season (Table 3 and Table 5). Thus, RR had higher annual NEEB than DR (Table 5), which can be primarily attributed to the high grain yield in the main season and lower total costs in the ratoon season. The current rice grain yield of the ratoon season is up to 7.56 t ha^−1^ in Hubei Province under optimal management and appropriate growth conditions (suitable ambient temperature, soil, rainfall, etc.) [48]. Therefore, these results suggest that RR is a sustainable cropping system with a lower greenhouse effect and higher NEEB.

In summary, our study found that RR and DR, two widely adopted rice cropping systems in China, had great differences in GHG emissions, GWP, GHGI, and NEEB. Compared with DR, RR had lower annual GWP and GHGI, and higher annual NEEB. It is noteworthy that our study also compared the seasonal differences in the four parameters between two seasons in both DR and RR and between the late season in DR and the ratoon season in RR to exclude to a great extent the effects of environmental factors. In addition, our study also investigated the associations of LAI, biomass, and grain yield with GHG emissions in the two seasons of RR and DR. Therefore, our study provides an understanding of GHG emissions and NEEB and further illuminates with previous studies together the characteristics of good annual grain yields and low carbon emissions in RR, which is increasingly adopted in the middle reach of the Yangtze River in China.

## 4. Materials and Methods

### 4.1. Field Site and Soil Characteristics

The field experiment was conducted in 2018 and 2019 at farm fields in Huaqiao town, Wuxue city, Hubei province (30°00′ N, 115°44′ E). The region is in the middle reaches of the Yangtze river and has a subtropical monsoonal climate. The accumulated temperature, accumulated radiation, and precipitation in the rice growing season are shown in Table 6. For convenience, the main season of RR and early season of DR were referred to as the first season, and the ratoon season of RR and late season of DR were referred to as the second season.

The basic characteristics of the 0–20 cm soil layer at the experimental fields were as follows: pH 6.5, organic carbon content 28.04 g kg^−1^, total nitrogen content 3.24 g kg^−1^, total phosphorus content 0.66 g kg^−1^, total potassium content 10.93 g kg^−1^, available nitrogen content 123.6 mg kg^−1^, available phosphorus content 8.26 mg kg^−1^, and available potassium content 158.44 mg kg^−1^. The experimental field was planted with rice in 2017 and then left fallow from November 2017 to April 2018.

### 4.2. Experimental Treatments and Management Practices

The two cropping systems (DR and RR) were established in a randomized complete block design with four replicates, and the area of each plot was 64 m^2^ (8 m × 8 m). The widely planted indica rice variety Liang You 6326 (LY6326) was used in RR, which has been proven to be suitable for RR in central China [49]. In DR, the indica rice variety Liang You 287 (LY287) was used for the early season because of its relatively short whole growth duration and relatively high yield, and the indica rice variety Huang Hua Zhan (HHZ), which has been planted in a large area in central and southern China [50], was used for the late season due to its higher yield and good rice quality. The performances of rice plants at tillering and maturity stages in the two seasons in RR and DR are shown in Appendix A.

The fertilizer application in each season is shown in Table 7. In the main season of RR, nitrogen fertilizer (urea, 46.4% N) was applied in three splits (basal fertilizer, tiller fertilizer, and panicle fertilizer) at a ratio of 5:2:3. All phosphate fertilizer (calcium superphosphate, with 12% phosphorus pentoxide) was used as basal fertilizer, and potassium fertilizer (potassium chloride, with 60% potassium oxide) was applied in two splits (basal fertilizer and panicle fertilizer) at a ratio of 3:2. In the ratoon season, nitrogen fertilizer was topdressed twice (bud-promoting fertilizer on the 10th day after the heading stage in the main season and tiller-promoting fertilizer on the 2nd day after harvest of the main crops) at a ratio of 1:1. Potassium fertilizer was also applied as bud-promoting fertilizer.

In the early season of DR, nitrogen fertilizer was applied in three splits at a ratio of 2:2:1 (basal fertilizer, tiller fertilizer, and panicle fertilizer). Phosphate fertilizer was used as the basal fertilizer, and potassium fertilizer was applied in two splits (basal fertilizer and panicle fertilizer) at a ratio of 4:3. In the late season, nitrogen fertilizer was applied in three splits at a ratio of 4:3:3 (basal fertilizer, tiller fertilizer, and panicle fertilizer); phosphate fertilizer was used as the basal fertilizer; and potassium fertilizer was applied in two splits (basal fertilizer and panicle fertilizer) at a ratio of 3:2.

Rice seedlings were raised for 30 days before transplanting. The transplantation and harvest dates are shown in Table 6. In RR, tillage was performed before transplanting in the main crop season at a planting spacing of 13.3 × 30 cm, with two seedlings per hill, and rice stubbles of 40 cm were retained when the main crops were harvested. In DR, tillage was performed before transplanting in both seasons, with a planting spacing of 13.3 × 20 cm and two seedlings per hill. The planting spacing followed local high-yielding management. At the mature stage in the early season, the plants were harvested at 40 cm from the ground; then, the stubbles were plowed into soils and flooded to promote the decay of the stubbles, followed by seedling transplanting.

In the early and late seasons of DR and the main season of RR, a 3–5 cm water layer was maintained after transplanting. Field drying was performed for about 7 days at the maximum tillering stage and then the field was re-irrigated, followed by alternate wetting and drying treatment until 7 days before harvest at maturity. In the ratoon season, for application of tiller-promoting fertilizer and the growth of regenerated buds, a 3–5 cm water layer was maintained for approximately 7 days after the harvest of the main season crops, then alternate wetting and drying treatment was also performed until approximately 7 days before harvest. The control of pests, diseases, and weeds was performed for high-yielding practices.

### 4.3. Collection and Measurement of CH_4_ and N_2_O

CH_4_ and N_2_O were collected using the static chamber method [51]. The chamber consisted of a PVC pipe with a 30 cm diameter. A digital electronic thermometer was inserted into the chamber from the top to monitor the real-time temperature inside the chamber. A fan with a lithium-ion battery (Jinhongrun Electronic Factory, Shenzhen, China) was installed on the inside top of the chamber to fully mix the gases inside. A 9-mm diameter glass tube was inserted in the middle of the chamber, and the end outside of the chamber was connected to a plastic three-way valve through a rubber tube for gas sampling. The heights of the chambers were 60 and 120 cm, which were used when the plant height was lower and higher than 50 cm, respectively.

Successional collection of gases was performed from 8:00 to 12:00 after transplanting once a week. A 30-mL syringe with a three-way valve at the top was used to store the sampled gases. Before gas collection, the chamber was placed on a circular stainless steel base (Hengda Hardware, Wuxue, China) pre-buried in soil, one for each plot, and the circular base was filled with water for sealing. At gas sampling, the three-way valve attached to the syringe was connected to the three-way valve of the chamber, and 30 mL gas was sampled from each chamber at 0, 10, 20 and 30 min after the chamber was sealed. The concentrations of CH_4_ and N_2_O in gas samples were measured using a gas chromatography system (GC-2010 plus, Shimadzu, Japan). The CH_4_ and N_2_O fluxes were calculated according to the following equation [52].
F = ρ × h × dc/dt × 273/(273 + T)(1)
where F is the gas emission flux (mg m^−2^ h^−1^ for CH_4_, μg m^−2^ h^−1^ for N_2_O), ρ is the density of the gas under standard conditions (0.714 kg m^−3^ for CH_4_ and 1.964 kg m^−3^ for N_2_O), h is the height of the chamber (m), T is the average temperature inside the chamber (°C) during gas collection, and dc/dt is the change rate of gas concentration inside the chamber per unit time (ppm min^−1^).

Cumulative CH_4_ and N_2_O emissions were calculated as follows:CE = ∑[(F_i_ + F_i + 1_)/2 × 24 × d](2)
where CE is the cumulative gas emissions, F_i_ and F_i + 1_ are the gas emission fluxes at two consecutive adjacent sampling time points, and d is the number of days for the interval.

### 4.4. Sampling of Plants and Yield Measurement

At the heading stage, 12 uniform rice plants were selected from each plot and divided into stems, leaves, and panicles. A desktop leaf area meter was used to measure leaf area and calculate the leaf area index (defined as the leaf area per unit of land area, LAI). Then, the stems, leaves, and panicles were oven-dried at 80 °C to a constant weight and the total biomass accumulation was determined. At the maturity stage, 12 uniform rice plants were selected from each plot for the biomass of aboveground plants. The biomass accumulation from the heading stage to the maturity stage (BAH) was the difference between the two stages.

The rice yields were measured in an area of 5 m^2^ per plot, and the grain was weighed after drying in the sun and converted to yield at a moisture content of 14%.

### 4.5. GWP and GHGI

The global warming potential (GWP, t CO_2_-eq ha^−1^) was calculated as follows [43]:GWP = CE(CH_4_) × 34 + CE(N_2_O) × 298(3)
where CE (CH_4_) and CE (N_2_O) represent the cumulative emissions of CH_4_ and N_2_O, respectively, and 34 and 298 are coefficients used to convert the cumulative emissions of CH_4_ and N_2_O into CO_2_ equivalent emissions.

The GHG intensity-based grain yield (GHGI, t CO_2_-eq t^−1^ yield) was defined as the ratio of GWP to grain yield [43].

### 4.6. NEEB

The calculation of net ecosystem economic benefit (NEEB) was based on the following formula [24]:NEEB = GY _income_ − AI _cost_ − GWP _cost_(4)
where GY _income_ is the economic income based on grain yield and unit price of rice grains, AI _cost_ refers to the costs of agricultural inputs (seeds, fertilizers, pesticides, and labor), GWP _cost_ (carbon costs) is the product of GWP and carbon-trading price. In this study, the carbon-trading price was taken as 232.7 Chinese yuan (CNY) t^−1^ CO_2_-eq [24]. The total cost is the sum of AI _cost_ and GWP _cost_. The unit prices of agricultural materials [53], labor, carbon, and rice grains [54] are shown in Appendix A.

### 4.7. Statistical Analysis

The value of a given parameter was expressed as the mean of four replicates with standard error (SE) using the SigmaPlot 10.0 software package (SPSS Inc., Chicago, IL, USA). The difference significance of averages was evaluated using the least significant difference (LSD) test at a 5% probability level using the Statistix 9 software package (Analytical software, Tallahassee, FL, USA). Pearson correlation analysis was used to estimate the correlation coefficients among parameters.

## 5. Conclusions

RR had significantly lower annual cumulative CH_4_ emissions by 35.21% than DR due to the lower cumulative CH_4_ emissions in the ratoon season; however, the two cropping systems had similar annual cumulative N_2_O emissions. The leaf area index (LAI) and biomass accumulation in the ratoon season were significantly lower than those in the main season and late season. The three investigated plant traits (leaf area index, biomass accumulation after flowering, and aboveground biomass accumulation at maturity stage) had high positive correlations with paddy CH_4_ emissions in the second season, and the poorer performance of RR in the three traits could account for lower CH_4_ emissions by 73.23% in the ratoon season compared with those in the late season in DR. The annual GWP and GHGI of RR were significantly lower by 34.43 and 30.76% than those of DR, which could be primarily attributed to lower cumulative CH_4_ emissions. The lower costs of agricultural input and carbon cost could account for higher annual NEEB in RR relative to DR. This study shows that RR has significantly lower annual cumulative CH_4_ emissions, GWP, and GHGI and significantly higher annual NEEB relative to DR, and the annual grain yield is comparable. Therefore, our results suggest that RR is more environmentally friendly and sustainable than DR. Previous studies always indicate that the amount of CH_4_ emissions is determined by soil CH_4_ production and CH_4_ oxidation, and most paddy CH_4_ is emitted via rice plants. Therefore, future research should focus on the seasonal effects of rice plants on the soil CH_4_ production and oxidation in RR and DR, and investigate the effects of the morphological and anatomical characteristics of rice plants on paddy CH_4_ emissions and seasonal differences. 

## Figures and Tables

**Figure 1 plants-12-03354-f001:**
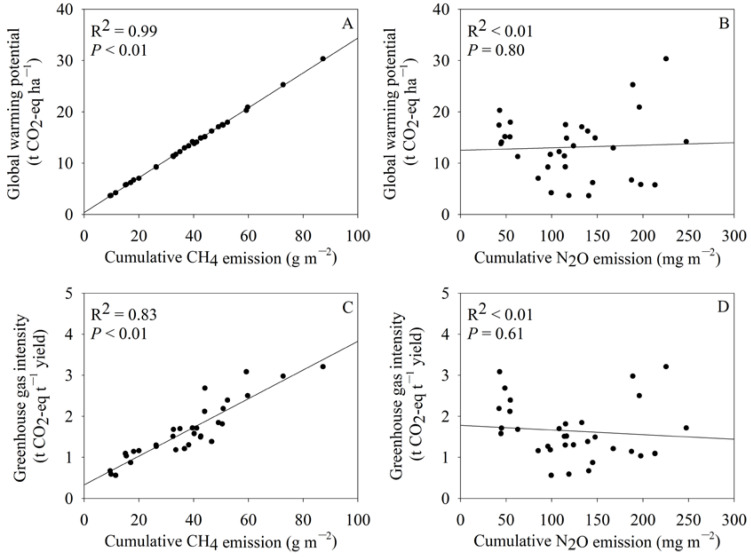
Relationships of global warming potential with cumulative CH_4_ emissions (**A**) and cumulative N_2_O emissions (**B**) and relationships of greenhouse gas intensity with cumulative CH_4_ emissions (**C**) and cumulative N_2_O emissions (**D**) across two seasons of the two cropping systems and the two years (n = 32).

**Table 1 plants-12-03354-t001:** Seasonal grain yields and aboveground characteristics in different crop systems.

Year	System	First Season					Second Season			
		LAI	BAH	Biomass	Yield		LAI	BAH	Biomass	Yield
		(m^2^ m^−2^)	(kg m^−2^)	(kg m^−2^)	(t ha^−1^)		(m^2^ m^−2^)	(kg m^−2^)	(kg m^−2^)	(t ha^−1^)
2018	DR	3.18 ± 0.33 b	0.74 ± 0.07 a	1.34 ± 0.06 b	7.29 ± 0.08 b		5.12 ± 0.19 a *	0.90 ± 0.11 a	2.01 ± 0.09 a *	8.64 ± 0.28 a *
	RR	3.94 ± 0.28 a	0.88 ± 0.11 a *	1.64 ± 0.08 a *	10.67 ± 0.39 a *		3.50 ± 0.26 b	0.24 ± 0.07 b	1.30 ± 0.03 b	5.55 ± 0.13 b
2019	DR	2.35 ± 0.19 b	0.63 ± 0.07 a	1.24 ± 0.06 b	6.75 ± 0.40 b		3.82 ± 0.57 a	0.57 ± 0.04 b	1.86 ± 0.02 a *	9.64 ± 0.14 a *
	RR	4.00 ± 0.17 a *	0.73 ± 0.06 a	1.96 ± 0.02 a *	7.88 ± 0.46 a		1.92 ± 0.22 b	0.78 ± 0.03 a	1.38 ± 0.01 b	6.73 ± 0.35 b
Average	DR	2.76 ± 0.22 b	0.68 ± 0.07 a	1.29 ± 0.05 b	7.02 ± 0.21 b		4.47 ± 0.33 a *	0.74 ± 0.05 a	1.93 ± 0.04 a *	9.14 ± 0.08 a *
	RR	3.97 ± 0.17 a *	0.80 ± 0.05 a *	1.80 ± 0.04 a *	9.27 ± 0.33 a *		2.71 ± 0.06 b	0.51 ± 0.05 b	1.34 ± 0.02 b	6.14 ± 0.23 b

Note: DR, double rice cropping system; RR, ratoon rice cropping system; first season refers to the early season of DR and main season of RR, second season refers to the late season of DR and ratoon season of RR. Different lowercase letters within the same column indicate significant differences (*p* < 0.05) between the two systems in the same season and year (average). * indicates significant differences (*p* < 0.05) between the two seasons in the same cropping system. LAI, leaf area index at full heading stage; BAH, biomass accumulation from the full heading stage to maturity stage; Biomass, plant aboveground biomass at maturity stage.

**Table 2 plants-12-03354-t002:** Cumulative CH_4_ (g m^−2^) and N_2_O (mg m^−2^) emissions in the rice growing season in different crop systems.

Year	System	First Season			Second Season			Annual	
		CH_4_	N_2_O		CH_4_	N_2_O		CH_4_	N_2_O
2018	DR	30.03 ± 2.21 b	108.18 ± 4.38 a		64.83 ± 10.12 a *	214.40 ± 13.58 a *		94.86 ± 12.17 a	322.58 ± 9.49 a
	RR	41.01 ± 2.24 a *	144.63 ± 9.06 a		14.52 ± 1.80 b	184.69 ± 15.62 a		55.53 ± 2.12 b	329.32 ± 15.68 a
2019	DR	43.30 ± 4.05 a	55.06 ± 2.92 a		43.92 ± 3.86 a	115.75 ± 7.03 a *		87.23 ± 4.17 a	170.81 ± 6.45 a
	RR	47.85 ± 4.50 a *	43.63 ± 0.64 b		14.59 ± 2.36 b	112.06 ± 12.94 a *		62.45 ± 5.23 b	155.70 ± 12.83 a
Average	DR	36.67 ± 2.41 a	81.62 ± 1.61 a		54.38 ± 4.91 a *	165.07 ± 9.14 a *		91.04 ± 6.79 a	246.69 ± 7.68 a
	RR	44.43 ± 1.55 a *	94.13 ± 4.45 a *		14.56 ± 0.77 b	148.37 ± 13.28 a *		58.99 ± 2.06 b	242.51 ± 11.40 a

Note: DR, double rice cropping system; RR, ratoon rice cropping system; first season refers to the early season of DR and main season of RR; second season refers to the late season of DR and ratoon season of RR. Different lowercase letters within the same column indicate significant differences (*p* < 0.05) between the two systems in the same season and year (average). * indicates significant differences (*p* < 0.05) between the two seasons in the same cropping system.

**Table 3 plants-12-03354-t003:** Warming potential (GWP, t CO_2_-eq ha^−1^) and greenhouse gas intensity based grain yields (GHGI, t CO_2_-eq t^−1^ yield) in the rice growing season in different crop systems.

Year	System	First Season			Second Season			Annual	
		GWP	GHGI		GWP	GHGI		GWP	GHGI
2018	DR	10.53 ± 0.76 b	1.44 ± 0.10 a		22.68 ± 3.42 a *	2.60 ± 0.33 a *		33.21 ± 4.13 a	2.07 ± 0.23 a
	RR	14.37 ± 0.75 a *	1.35 ± 0.06 a		5.49 ± 0.65 b	0.99 ± 0.11 b		19.86 ± 0.76 b	1.22 ± 0.02 b
2019	DR	14.89 ± 1.37 a	2.22 ± 0.21 a		15.28 ± 1.33 a	1.59 ± 0.15 a		30.17 ± 1.43 a	1.85 ± 0.12 a
	RR	16.40 ± 1.53 a *	2.14 ± 0.34 a *		5.30 ± 0.80 b	0.80 ± 0.14 b		21.70 ± 1.80 b	1.49 ± 0.15 b
Average	DR	12.71 ± 0.82 a	1.83 ± 0.15 a		18.98 ± 1.67 a *	2.10 ± 0.16 a *		31.69 ± 2.31 a	1.96 ± 0.15 a
	RR	15.39 ± 0.52 a *	1.74 ± 0.15 a *		5.39 ± 0.26 b	0.89 ± 0.03 b		20.78 ± 0.72 b	1.36 ± 0.07 b

Note: DR, double rice cropping system; RR, ratoon rice cropping system; first season refers to the early season of DR and main season of RR; and the second season refers to the late season of DR and ratoon season of RR. Different lowercase letters within the same column indicate significant differences (*p* < 0.05) between systems in the same season and year (average). * indicates significant differences (*p* < 0.05) between the two seasons in the same cropping system.

**Table 4 plants-12-03354-t004:** Pearson’s correlations of cumulative CH_4_ emissions, warming potential (GWP) and greenhouse gas intensity based on grain yields (GHGI) with aboveground characteristics and grain yield in the first season and second season.

Season		LAI	BAH	Biomass	Grain Yield
First season	CH_4_	−0.05	0.16	0.47	0.03
	GWP	−0.04	0.18	0.47	0.06
	GHGI	−0.36	−0.24	0.06	−0.60 *
Second season	CH_4_	0.60 *	0.51 *	0.89 **	0.74 **
	GWP	0.61 *	0.50 *	0.88 **	0.74 **
	GHGI	0.61 *	0.49	0.83 **	0.60 *

Note: * indicates significance at the 0.05 probability level, ** indicates significance at the 0.01 probability level. LAI, leaf area index; BAH, biomass accumulation from the full heading stage to maturity stage; Biomass, aboveground biomass of maturity stage (*n* = 16).

**Table 5 plants-12-03354-t005:** Grain incomes, total costs, and net ecosystem economic benefits (× 10^3^ CNY ha^−1^) in the rice growing season in different crop systems.

Year	System	First Season				Second Season				Annual		
		Grain Incomes	Total Costs	Net Ecosystem Economic Benefits		Grain Incomes	Total Costs	Net Ecosystem Economic Benefits		Grain Incomes	Total Costs	Net Ecosystem Economic Benefits
2018	DR	19.70 ± 0.22 b	12.60 ± 0.18 a	7.10 ± 0.24 b		25.93 ± 0.83 a *	15.55 ± 0.80 a *	10.83 ± 0.46 b *		45.63 ± 0.80 a	28.15 ± 0.96 a	17.48 ± 0.42 b
	RR	28.80 ± 1.05 a *	13.20 ± 0.18 a *	15.60 ± 0.96 a		16.64 ± 0.38 b	4.00 ± 0.15 b	12.64 ± 0.32 a		45.44 ± 0.96 a	17.21 ± 0.18 b	28.23 ± 0.81 a
2019	DR	18.23 ± 1.09 b	13.61 ± 0.32 a	4.62 ± 1.03 b		28.91 ± 0.42 a *	13.83 ± 0.31 a	15.09 ± 0.68 a *		47.14 ± 1.50 a	27.44 ± 0.33 a	19.70 ± 1.63 b
	RR	21.27 ± 1.25 a	13.67 ± 0.36 a *	7.60 ± 1.60 a		20.20 ± 1.04 b	3.96 ± 0.19 b	16.25 ± 1.10 a *		41.47 ± 1.02 b	17.63 ± 0.42 b	23.84 ± 1.26 a
Average	DR	18.96 ± 0.58 b	13.11 ± 0.19 a	5.86 ± 0.62 b		27.42 ± 0.25 a *	14.69 ± 0.39 a *	12.73 ± 0.16 a *		46.38 ± 0.47 a	27.79 ± 0.54 a	18.59 ± 0.78 b
	RR	25.04 ± 0.88 a *	13.44 ± 0.12 a *	11.60 ± 0.97 a		18.42 ± 0.70 b	3.98 ± 0.06 b	14.43 ± 0.66 a		43.46 ± 0.78 b	17.42 ± 0.17 b	26.04 ± 0.88 a

Note: DR, double rice cropping system; RR, ratoon rice cropping system. The first season refers to early season of DR and main season of RR; the second season refers to the late season of DR and ratoon season of RR. The total costs refer to the sum of the costs of agricultural inputs and carbon costs (the product of GWP and carbon-trading price), and the net ecosystem economic benefits refer to the grain incomes minus the total cost. Different lowercase letters within the same column indicate significant differences (*p* < 0.05) between the two systems in the same season and year (average). * indicates significant differences (*p* < 0.05) between the two seasons in the same cropping system.

**Table 6 plants-12-03354-t006:** Date of transplanting and harvest, growth duration, cumulative temperature, solar radiation and precipitation.

Year	Season	System	Transplanting Date	Harvest Date	Growth Duration	Cumulative Temperature	Cumulative Solar Radiation	Precipitation
			(mm/dd)	(mm/dd)	(d)	(°C)	(MJ m^−2^)	(mm)
2018	First season	DR	04/27	07/10	75	1906	1158	312
		RR	04/27	08/07	103	2747	1778	341
	Second season	DR	07/18	10/24	99	2486	1650	119
		RR	–	10/06	60	1547	996	60
2019	First season	DR	04/27	07/22	87	1963	1052	462
		RR	04/27	08/14	110	2630	1532	463
	Second season	DR	07/25	10/28	96	2407	1605	12
		RR	–	10/24	71	1408	963	10

Note: DR, double rice cropping system; RR, ratoon rice cropping system; first season refers to the early season of DR and main season of RR; second season refers to late season of DR and ratoon season of RR.

**Table 7 plants-12-03354-t007:** Inputs applied to crop production under different cropping systems.

Season	System	Fertilizers (kg ha^−1^)				Pesticides (kg ha^−1^)			Seeds	Labor (Person ha^−1^)				
		N	P	K		Herbicides	Insecticides	Fungicides	(kg ha^−1^)	Seedling Raise	Transplanting	Harvest	Tillage	Water Management
First season	DR	150	40	70		0.375	3.6	0.45	37.5	1	15	15	30	1
	RR	200	40	100		0.375	3.6	0.45	22.5	1	15	15	30	1
Second season	DR	150	40	100		0.375	3.6	0.45	37.5	1	15	15	30	1
	RR	150	0	50		0	0	0	0	0	0	15	0	0.5

Note: DR, double rice cropping system; RR, ratoon rice cropping system; first season refers to the early season of DR and main season of RR; second season refers to the late season of DR and ratoon season of RR. The inputs applied to crop production were the same in 2018 and 2019.

## Data Availability

All data supporting the findings of this study are available within the paper and its online Appendix A.

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
