# Peer review of "Ratoon Rice Cropping Mitigates the Greenhouse Effect by Reducing CH4 Emissions through Reduction of Biomass during the Ratoon Season"

_plants, 2023, doi:10.3390/plants12193354_

Round 1

Reviewer 1 Report

Report on plants-2558580

This is a well written manuscript, with a lot of interesting data. However, most of what is reported is already known. The title mentions ‘reduction of biomass during the ratoon season’, but the abstract barely refers to this, likewise the Conclusions do not play this up.

As I see from the data, the differences between DR and RR lie in the relative durations of the first and second crops in the sequence. The authors do not dwell on this, and I wonder how the results would be if the DR first crop were much longer and the RR first crop much shorter than in the current trial? Would this affect the beneficial outcome of the RR over the DR? For all the data collected, there is not much that is not known already, so I recommend the authors discuss in some more detail their interpretations of the data, and less on the repetitive/confirmation nature of much of the data.

As mentioned, the presentation is excellent [as is the literature cited, all very up to date], so I have only had to annotate on a few occasions where there is some confusion, or query.

In the Introduction, I note ‘so…’ on  a few occasions, since the authors do not really help the reader to come  to some sort of conclusion.

Line 98-99 what about references 16 and 19?

In the Tables, why is 2018 underlined and in bold?

In Table 4, why the contrary correlations between grain yield and GHGI in the first and second season?

Lines 195 onwards, this simply repeats the result of yields!!s So is it really useful to present these data?

Table 5 and elsewhere what is CNY?

Lines 288-298 are essentially repeats of lines 228-243!

Line 340 better growth conditions than what??

Lines 394-396 why was there no 3-5 cm water layer imposed as in the late season of DR?

Line 456 how were the carbon trading prices derived?

Very good.

Author Response

Dear reviewer 1:

Thank you for constructive comments on our manuscript. We have carefully considered your comments and suggestions and made some changes.

Please find our responses below. To make it easier for you to review, your original comments and suggestions are displayed in italics in each serial number, and our responses are displayed after “Authors Response”.

Thank you again for your advice, and we hope to learn more from you!

  1. The title mentions ‘reduction of biomass during the ratoon season’, but the abstract barely refers to this, likewise the Conclusions do not play this up.

Authors Response: Thank you for your comments! In our revised manuscript, we added the characteristics of leaf area index and biomass accumulation of rice plants in the ratoon season and its effect on methane emission in the Abstract section:

The leaf area index (LAI) and biomass accumulation in the ratoon season were significantly lower than those in the main season and late season, and CH4 emissions, GWP, and GHGI showed significant positive correlations with LAI, biomass accumulation and grain yield in the ratoon and late season”.

In the Conclusion, we also emphasized the effect of plant biomass accumulation in the ratoon season on methane emissions: The three investigated plant traits (leaf area index, biomass accumulation after flowering, and aboveground biomass accumulation at maturity stage) had high positive correlations with paddy CH4 emissions in the ratoon and late seasons, and the poorer performance of RR in the three traits may associate with the lower CH4 emissions in the ratoon season compared with those in the late season in DR”.

  1. As I see from the data, the differences between DR and RR lie in the relative durations of the first and second crops in the sequence. The authors do not dwell on this, and I wonder how the results would be if the DR first crop were much longer and the RR first crop much shorter than in the current trial? Would this affect the beneficial outcome of the RR over the DR? For all the data collected, there is not much that is not known already, so I recommend the authors discuss in some more detail their interpretations of the data, and less on the repetitive/confirmation nature of much of the data.

Authors Response: Thank you for your comments. This is an excellent suggestion. If the growth duration is extended, more temperature and radiation resources will be available for rice growth, which may increase rice yield. Currently, ratoon rice is recommended to grow in the region, in which thermal energy is more than that required for single rice but not enough for planting double rice (Dong et al., 2017; Xu et al., 2021). In addition, rice varieties with appropriate duration are also selected based on local climatic resources, as well as the appropriate dates of sowing and transplanting for the main season of ratoon rice are determined accordingly (Xu et al., 2021; Yu et al., 2022; Hu et al., 2023; Zhang et al., 2023).

In central China, transplanting of the main season is usually performed on approximately April 20, and the harvesting of the main season is usually performed during August 5 and 15 (Peng et al., 2023). However, if the duration of the main season is much shorter than that of current varieties commonly adopted in ratoon rice practice, the main crop should be transplanted early, resulting in the danger for meeting low temperature in early April. In this case, the ratoon season plants are often at full heading and grain filling stages at August 10 or even earlier, during which the temperature is often stressful high and is not conducive to the grain filling of the ratoon season, resulting in reduction of the ratoon season rice yield. Therefore, in ratoon rice production, the growth duration of the main season must be appropriate in order to ensure the safety for full heading of rice plants in the ratoon season and sufficient temperature and radiation resources for ratoon crop growth (Yu et al., 2022).

Similarly, in the double rice system, the growth duration of the early rice influences the growth of the late rice. For example, if the duration of the early rice is extended, then the late rice was transplanted accordingly, the case may increase the potential risk of low temperature when the late season rice is in full heading and grain filling, thus reducing rice yield in the late season. Therefore, in rice planting area suitable for double rice systems in China, rice varieties with a relatively short growth duration are generally adopted for the early rice season, which enables both early- and late-season rice to fully utilize temperature and radiation resources and to obtain a higher annual yield. In general, both the double rice system and ratoon rice system consist of two rice growing seasons, and farmers are more concerned with annual yields.

In fact, we do not really know how the results would be if the DR first crop were much longer and the RR first crop much shorter than in the current trial. Because CH4 emissions are often affected by several factors, such as growth duration, environmental conditions (ambient temperature, rainfall etc), crop growth status (affected by many factors). If the duration is changed, growth natural conditions are surely changed accordingly. Your comment is a wonderful scientific question and gives us a great idea for further research.

Noticeably, in our study presented in the manuscript, the varieties and farm management (seeding, transplanting, harvest, etc.) used in the trail are widely adopted in the double rice and ratoon rice practice for high annual grain yield and safe production based the local environments.

As response to the suggestion, we added the following several words in Discussion 3.4:

 “The NEEB of a given system is often calculated based the economic income (grain yield and unit price of rice grains), the costs of agricultural inputs (seeds, fertilizers, pesticides, and labor), and GWP cost (GWP and carbon-trading price) [24]. Therefore, when estimating the NEEB, several factors should be considered, such as variety characteristics, farm managements, etc. In the study, the crop characteristics and CH4 emissions were investigated in farmer fields, and the varieties and farm managements (seedling raising, transplanting, fertilization, irrigation, pest and disease control, and harvest) in the experiment are widely adopted in the double rice and ratoon rice practice for better annual grain yield and safe production based the local environments. So, the estimated NEEB in the study should reflect the fact in experimental site.”

References to this review are listed below:

(1) Dong HL, Chen Q, Wang WQ, Peng SB, Huang JL, Cui KH, Nie LX. The growth and yield of a wet-seeded rice-ratoon rice system in central China. Field Crops Research, 2017, 208: 55-59.

(2) Xu FX, Zhang L, Zhou XB, Guo XY, Zhu YC, Liu M, Xiong H, Jiang P. The ratoon rice system with high yield and high efficiency in China: progress, trend of theory and technology. Field Crops Research, 2021, 272: 108282.

(3) Yu X, Tao X, Liao J, Liu SC, Xu L, Yuan S, Zhang ZL, Wang F, Deng NY, Huang JL, Peng SB. Predicting potential cultivation region and paddy area for ratoon rice production in China using Maxent model. Field Crops Research, 2022, 275: 108372.

(4) Hu XY, Ma MJ, Huang ZB, Wu ZJ, Su BF, Wen ZH, Fu YQ, Pan JF, Liu YZ, Hu R, Li MJ, Liang KM, Zhong XH. Progress and challenges of rice ratooning technology in Guangdong Province, China. Crop and Environment, 2023, 2: 17-23.

(5) Zhang Q, Liu XC, Yu GL, Zhao HY, Feng DQ, Gu MX, Zhu T, Kuang X, Li BB. Progress and challenges of rice ratooning technology in the south of Henan Province, China. Crop and Environment, 2023, 2: 75-80.

(6) Peng SB, Zheng C, Yu X. Progress and challenges of rice ratooning technology in China. Crop and Environment, 2023, 2: 5-11.

  1. In the Introduction, I note ‘so…’ on a few occasions, since the authors do not really help the reader to come to some sort of conclusion.

Authors Response: Thank you for your comment. This is a very important comment for us to improve the Introduction section. We found that where you marked 'so...' we presented some results from references but did not summarize them. Therefore, we added a summary of the cited literature in our revision to help readers understand conclusion of previous studies. Please see the detail in the revised MS.

  1. Line 98-99 what about references 16 and 19?

Authors Response: Thank you for your comments. In your mentioned Line 98-99, we would like to point out that, in ratoon rice system (RR) and double rice system (DR), there were relatively few studies have been done on the effect of rice plant characteristics on methane emissions. Reference 16 comprehensively explored the effects of rice plant characteristics of super rice varieties as middle rice on paddy methane emissions, but the study was conducted in a single-season rice system and did not reflect the methane emission characteristics of RR and DR. Reference 19 mainly studied the effect of no-tillage on GHGs in rice fields and did not address the effect of rice plant characteristics on methane emissions. Based on our understanding, we have cited references 3 and 25 in the revised MS.

  1. In the Tables, why is 2018 underlined and in bold?

Authors Response: Thank you for pointing out carelessness. We noticed that this cases occurred in the PDF versions of Table 1, Table 2, Table 3, Table 5, and Table 7. Some system errors may have occurred when the editor converted our Word version of the manuscript to a PDF version. Therefore, we rechecked our uploaded Word manuscripts and standardized the formatting of the tables again. We will continue to monitor the format of tables and figures and actively communicate with editors in subsequent processes.

  1. In Table 4, why the contrary correlations between grain yield and GHGI in the first and second season?

Authors’ Response: Thank you for your comment. GHGI is the ratio of GWP to grain yield. In our study, there was no significant difference in GWP between early season of double rice and the main season of the ratoon rice. In this case, the impact of yield on GHGI may be greater than that of GWP, and high yield reduce GHGI. Therefore, in the first season, GHGI is negatively correlated with yield. This is similar to the results of Zhang et al. (2023).

In some previous studies, higher yields did not necessarily result in lower GHGI. For example, in the study of Gao et al. (2023) on maize, the increased yield resulted in significant increases of GHGI in several treatments. In our study, in the second season, the GWP of the late rice season was 3.52 times higher than that of the ratoon season, while the yield of the late season was 1.49 times higher than that of the ratoon season. It is possible that GWP rather than grain yield have a greater impact on GHGI, therefore, increased grain yield does not certainly lead to a lower GHGI, GHGI showed a positive correlation with yield. This results is in line with the results of Zhou et al. (2023). Our results suggest that the factors affecting seasonal GHGI in the late season and the ratoon season of ratoon rice may be different from that in the early rice and the main season. The detail for the differences deserves to be further explored.

Considering we did not discuss the related topic in the manuscript, so the related explanation did not add in the revised MS.

References to this review are listed below:

(1) Zhang GB, Yang YT, Wei ZJ, Zhu XL, Shen WY, Ma J, Lv SH, Xu H. The low greenhouse gas emission intensity in water-saving and drought-resistance rice in a rainfed paddy field in Southwest China. Field Crops Research, 2023, 302: 109045.

(2) Gao ZZ, Zhao JC, Wang C, Wang YX, Shang MF, Zhang ZP, Chen F, Chu QQ. A six-year record of greenhouse gas emissions in different growth stages of summer maize influenced by irrigation and nitrogen management. Field Crops Research, 2023, 290: 108744.

(3) Zhou YJ, Ji YL, Zhang M, Xu YZ, Li Z, Tu DB, Wu WG. Exploring a sustainable rice-cropping system to balance grain yield, environmental footprint and economic benefits in the middle and lower reaches of the Yangtze River in China. Journal of Cleaner Production, 2023, 404: 136988.

  1. Lines 195 onwards, this simply repeats the result of yields!! So is it really useful to present these data?

Authors Response: Thank you for your comments. In the study, economic benefit was calculated based grain yield and price per unit grains. Due to the price per unit grains of DR and RR were set as same, so economic benefit was positively associated with the grain yield.

In the section (2.5 Economic Benefits, Total Costs, and NEEB), we presented the grain incomes, total costs, and net ecosystem economic benefits. Farmers may pay more concerns on the costs and economic income of the cropping systems. The selection of a suitable cropping system is primarily dependent on high net economic benefits, which are mainly determined by the economic income of grain yields and costs of agricultural inputs. The three parameters (the grain incomes, total costs, and net ecosystem economic benefits) are also considered as cropping system selection for a given area by farmers and agricultural policy makers. Therefore, these data should be useful.

  1. Table 5 and elsewhere what is CNY?

Authors Response: Thank you for your comment. CNY stands for Chinese Yuan, the only standardized symbol for Chinese currency in international trade. We find that CNY is also used as a unit of economic quantity in some previous studies (Song et al., 2021; Xu et al., 2022). To avoid confusion for readers, we have added an explanation of CNY in the Materials and Methods section (4.6 NEEB) of the manuscript.

References to this review are listed below:

(1) Song KF, Zhang GB, Yu HY, Huang Q, Zhu XL, Wang TY, Xu H, Lv SH, Ma J. Evaluation of methane and nitrous oxide emissions in a three-year case study on single rice and ratoon rice paddy fields. Journal of Cleaner Production, 2021, 297: 126650.

(2) Xu Y, Liang LQ, Wang BR, Xiang JB, Gao MT, Fu ZQ, Long P, Luo HB, Huang C. Conversion from double-season rice to ratoon rice paddy fields reduces carbon footprint and enhances net ecosystem economic benefit. Science of the Total Environment, 2022, 813: 152550.

  1. Lines 288-298 are essentially repeats of lines 228-243!

Authors Response: Thank you for your comment. Lines 228-243 discussed the differences in methane emissions between the late season and ratoon seasons, Line 288-298 mainly focused on the differences in methane emissions between the main and ratoon seasons in ratoon rice cropping system. LAI, biomass, and root exudates were associated with the methane emissions during the late season, main season and ratoon season. So, when discussing, we also used LAI and biomass. For reducing repeat, we revised the statement.

  1. Line 340 better growth conditions than what??

Authors Response: Thank you for your comment. What we want to express as 'better growth conditions' are the growth conditions without weather disasters (such as high temperature, low temperature, and heavy rainfall). In the revised MS, we changed the words to “appropriate growth conditions (suitable ambient temperature, soil, rainfall, etc.)”

  1. Lines 394-396 why was there no 3-5 cm water layer imposed as in the late season of DR?

Authors Response: Thank you for your comment. In practices, it is necessary to maintain a water layer of 3-5 cm after transplanting to promote the growth of rice seedlings. We introduced water management (a water layer of 3-5 cm) for the early and late rice seasons of DR, as well as the main season of RR. In these three seasons (including the late rice season of DR you mentioned).

 Your question may be why the ratoon season of RR did not maintain a water layer of 3-5 cm. In fact, after rice harvesting in the main season crops, it is also necessary to maintain a water layer for application of tiller-promoting fertilizer. We have added related information on water management during the ratoon season in revised MS.

  1. Line 456 how were the carbon trading prices derived?

Authors Response: Thank you for your comment. For carbon trading prices, we cited a previous study in Section 4.6 for NEEB calculation and Table S1 (Song et al., 2021).

References to this review are listed below:

(1) Song KF, Zhang GB, Yu HY, Huang Q, Zhu XL, Wang TY, Xu H, Lv SH, Ma J. Evaluation of methane and nitrous oxide emissions in a three-year case study on single rice and ratoon rice paddy fields. Journal of Cleaner Production, 2021, 297: 126650.

Reviewer 2 Report

Reviewed article titled „Ratoon Rice Cropping Mitigates Global Warming Potential and Greenhouse Gas Intensity by Reducing CH4 Emissions Through Reduction of Biomass During the Ratoon Season” deals with a very important global issue. I read it with interest, although reading it requires great concentration.

The article is written very professionally and could be accepted for publication without changes. However, I propose to consider some of my suggestions:

1. At the end of the Discussion, it is worth referring to the Introduction and clearly emphasizing what is new in this work compared to previously published data.

2. The word "lower" is used many times in the Conclusions. After this word, it would be worth expressing it in % (specify in brackets how much % the given feature was lower).

3. In the Conclusions, the authors should indicate the further direction of research.

4. It would be very interesting for readers (especially for those outside of Asia) if the Authors could add some photos from the experimental fields.

Other minor suggestions:

Keywords should be different from the words in the title of the article.

Line 107: LAI and BAH  -  these words are used here for the first time and therefore require explanations (despite the appropriate explanations under the tables).

Below most of the tables is the following information “* indicates significant differences (P < 0.05) between the two seasons in the same cropping system and year (average)”. I think that it should be: “* indicates significant differences (P < 0.05) between the two seasons in the same cropping system”.

Lines 385 and 388: Authors should explain why the planting spacing was different.

Author Response

Dear reviewer 2:

Thank you for your decision and constructive comments on my manuscript. We have carefully considered your comments and suggestions and made some changes.

Please find our responses below. To make it easier for you to review, your original comments and suggestions are displayed in italics in each serial number, and our responses are displayed after “Authors Response”.

  1. At the end of the Discussion, it is worth referring to the Introduction and clearly emphasizing what is new in this work compared to previously published data.

Authors Response: Thank you for your comment. We added a paragraph at the end of the Discussion. In this paragraph, we summarize our results and respond to the issues mentioned in the introduction. In addition, we also emphasized what is new compared to previous studies. The added summary is as follows:

In summary, our study found that RR and DR, two widely adopted rice cropping systems in China, had great differences in GHG emissions, GWP, GHGI, and NEEB.  Compared with DR, RR had lower annual GWP and GHGI, and higher annual NEEB. It is noteworthy that our study also compared the seasonal differences in the four parameters between two seasons in both DR and RR, and between the late season in DR and the ratoon season in RR for excluding to a great extent the effects of environmental factors. In addition, our study also investigated the associations of LAI, biomass, and grain yield with GHG emissions in the two seasons of RR and DR. Therefore, our study provides an understanding for GHG emissions and NEEB, and further illuminates with previous studies together the characteristics of good annual grain yields and low carbon emissions in RR, which is increasingly adopted in the middle reach of the Yangtze River in China.

  1. The word "lower" is used many times in the Conclusions. After this word, it would be worth expressing it in % (specify in brackets how much % the given feature was lower).

Authors Response: Thank you for your comment. Based on your suggestion, we have made modifications to the conclusion.

  1. In the Conclusions, the authors should indicate the further direction of research.

Authors Response: Thank you for your comment. Based on your suggestion, we added the further direction of research at the end of the Conclusion as following:

The previous studies always indicate that amount of CH4 emissions is determined by soil CH4 production and CH4 oxidation, and most paddy CH4 is emitted via rice plants. Therefore, future research should focus on the seasonal effects of rice plants on the soil CH4 production and oxidation in RR and DR, and investigate the effects of the morphological and anatomical characteristics of rice plants on paddy CH4 emissions and seasonal differences.

  1. It would be very interesting for readers (especially for those outside of Asia) if the Authors could add some photos from the experimental fields.

Authors Response: Thank you for your comment. We added photos from the experimental fields. The plant performances of rice plants at the tillering and maturity stages in RR and DR were presented in Figure S5 of Supplementary Materials.

  1. Keywords should be different from the words in the title of the article.

Authors Response: Thank you for your comment. The current Keywords comprehensively reflect the main content of the article and may not need to be changed. In many published articles, global warming potential and greenhouse gas intensity were referred to as the greenhouse effect. Therefore, we changed the title of the article to “Ratoon rice cropping mitigates the greenhouse effect by reducing CH4 emissions through reduction of biomass during the ratoon season” to reduce repetition of keywords.

  1. Line 107: LAI and BAH - these words are used here for the first time and therefore require explanations (despite the appropriate explanations under the tables).

Authors Response: Thank you for your comment. Based on your suggestion, we added explanations of LAI and BAH when first appearance in Result section.

  1. Below most of the tables is the following information “* indicates significant differences (P < 0.05) between the two seasons in the same cropping system and year (average)”. I think that it should be: “* indicates significant differences (P < 0.05) between the two seasons in the same cropping system”.

Authors Response: Thank you for your comment. Based on your suggestion, we have changed accordingly in Table 1, Table 2, Table 3 and Table 5.

  1. Lines 385 and 388: Authors should explain why the planting spacing was different.

Authors Response: Thank you for your comment. As used for early rice, Liangyou 287 has shorter growth duration, and lower tillering ability and less panicle number per unit area than Liangyou 6326 that is often planted in ratoon rice system (Zheng et al., 2022; Cheng et al, 2011).

As a conventional rice variety, Huang Hua Zhan (planting as the late rice) also has lower tillering ability than Liangyou 6326. Therefore, to establishing high-yielding rice population, the planting density of the early rice and late rice was higher to compensate the low tillering ability and small panicle number in high-yielding cultivation practices. The high planting density is often used in double rice system in the middle China.

Liangyou 6326 is a super hybrid rice, with large biomass production capacity, several investigated have showed that high planting density often results several unexpected growth issues, such as lodging, more occurrence of diseases and pests, damage on regeneration buds, etc. In addition, when planting density reached 25 hill m-2 in ratoon rice, further increases in density did not increase, even decreased the rice yield of ratoon rice (Wang et al, 2023). Therefore, in our study, the planting spacing of the early season was 13.3 × 20 cm, while in the ratoon rice, the planting spacing was 13.3 × 30 cm. The planting spacing adopted in our study follows local high-yielding practices. We added explain about the difference of planting spacing in the revised MS.

References to this review are listed below:

(1) Zheng C, Wang YC, Yuan S, Xiao S, Sun YT, Huang JL, Peng SB. Heavy soil drying during mid-to-late grain filling stage of the main crop to reduce yield loss of the ratoon crop in a mechanized rice ratooning system. The Crop Journal, 2022, 10: 280-285.

(2) Cheng JP, Wu JP, Luo XW, Fan QZ, Zhang GZ, Zhou ZY, Zang Y, Yang BB. Influence of different planting methods on growth characteristics and yield of early rice. Hubei Agricultural Science, 2011, 50: 457-460. (In Chinese with English abstract).

(3) Xu FX, Zhang L, Zhou XB, Guo XY, Zhu YC, Liu M, Xiong H, Jiang P. The ratoon rice system with high yield and high efficiency in China: progress, trend of theory and technology. Field Crops Research, 2021, 272: 108282.

(4) Wang WQ, Zheng HB, Chen YW, Zou D, Luo YY, Tang QY. Progress and challenges of rice ratooning technology in Hunan Province, China. Crop and Environment, 2023, 2: 101-110.

Round 2

Reviewer 1 Report

See word file

See word file

Author Response

We  have revised the manuscript according to  yours suggestionson the second round review, and revised the terms according to your revision. we appreciate you very much for your carefull review and valuable suggestions.